# Malignant Pleural Mesothelioma Nodal Status: Where Are We at?

**DOI:** 10.3390/jcm10215177

**Published:** 2021-11-05

**Authors:** Sara Ricciardi, Francesco Carleo, Massimo O. Jaus, Marco Di Martino, Luigi Carbone, Alberto Ricci, Giuseppe Cardillo

**Affiliations:** 1Unity of Thoracic Surgery, Hospital of Bologna, IRCCS University, 40138 Bologna, Italy; 2Alma Mater, Studiorum University of Bologna, 40138 Bologna, Italy; 3Unity of Thoracic Surgery, San Camillo Forlanini Hospital, 00152 Rome, Italy; francesco.carleo@yahoo.it (F.C.); jausmassimo@inwind.it (M.O.J.); mdimartino@scamilloforlanini.rm.it (M.D.M.); lcarbone@scamilloforlanini.rm.it (L.C.); gcardillo@scamilloforlanini.rm.it (G.C.); 4Respiratory Unit, Sant’Andrea Hospital, 00189 Rome, Italy; alberto.ricci@uniroma1.it; 5Medicina Clinica e Molecolare, University of Rome La Sapienza, 00185 Rome, Italy

**Keywords:** malignant pleura mesothelioma, node positive MPM, staging, surgery, prognostic factors

## Abstract

Due to the lack of both prospective trial and high-volume retrospective studies, the management of clinical N+ malignant pleural mesothelioma (MPM) patients remains highly debated. Node positive patients show poor survival compared with node-negative ones; thus, lymph node staging appears crucial in determining treatment strategy. Notwithstanding the improvement in pre-treatment staging and the update on lymph node classification in the 8th edition of TNM, several open controversies remain on N parameter. How should we stage suspected N+ patients? How should we treat node positive patients? Which is the definition of a “resectable patient”? Is the site or the number the main prognostic factor for node positive patients? The aim of our narrative review is to analyse the existing relevant literature on lymph node status in MPM.

## 1. Introduction

Several open controversies remain on N parameter of Malignant Pleural Mesothelioma (MPM). Nodal involvement is recognised as a poor prognostic factor, but several weaknesses of nodal staging persist despite the improvement in the 8th edition of TNM classification [1]. N+ patients present poor survival compared to N− patients, therefore accurate lymph node staging is mandatory to plan a treatment strategy [2,3]. Besides, regarding both topography and extent, the N+ group is extremely heterogeneous [4]. According to ERS/ESTS/EACTS/ESTRO 2020 task force member, the use of non-invasive imaging is inaccurate in the assessment of nodal status, and even in case of direct biopsy, the presence of occult nodal metastasis cannot be excluded [5]. Moreover, endobronchial ultrasound (EBUS) and mediastinoscopy, useful techniques in clinical staging, are not able to detect nodal involvement in extra-mediastinal stations (e.g., internal mammary), peri-diaphragmatic or intercostal areas, frequently site of metastases in MPM. In absence of prospective randomized control trial, evidence-based approach including surgical therapy is debated. Based on data of high-volume centres, the N+ MPM disease has not been considered a contraindication to surgery. However, surgery alone seems not appropriate in node positive patients, and a multimodality approach, preferably in a context of a clinical trial, should be considered [6].

Multimodal treatment, including at least macroscopic complete resection and chemotherapy, is better than single modality in selected patients regarding OS, but it increases treatment-related morbidity and mortality [7]. Hence, clinicians should be conscious of the implications of the staging and treatment knowledge limitations when discussing with patients the pre-treatment prognosis.

## 2. Methods

Relevant literature searching was performed using keywords: “malignant pleural mesothelioma” and “lymphadenectomy” or “lymph node status” or “node positive MPM”. To identify the articles available in the literature, MEDLINE database was used and accessed through PubMed on August 2021.

The analysis was limited to full text version in English; case reports, unpublished papers and congress abstract were not included.

## 3. The Role of N Parameter

Lymph node involvement was first recognised as a negative prognostic factor in Malignant Pleural Mesothelioma (MPM) by Sugarbaker et al. in 1993 [8]. In MPM N+ patients, up to 50% reduction in long term survival has been described [9,10,11,12,13].

In this relentless local progressive disease, lymph node metastases are frequent: a recent analysis conducted by Mesothelioma Domain of IASLC Staging and Prognostic Factors Committee has shown frequent lymph node involvement (35–50%) in surgically treated patients, and a post-mortem study conducted by Rhian and co-workers describes nodal metastases in 76% of autopsies [14,15].

Although the pleural lymphatic drainage pathway is recognised clearly distinct than that of the lung, until 2018, nodal categories of MPM was inherited from lung cancer staging, due to lacking MPM-specific evidence. The 7th edition of the TNM, which utilised several pathological descriptors making it poorly applicable to non-surgical cohort and remained unchanged for about 20 years, classified ipsilateral nodes as either N1 (intrapleural, hilar and intraparenchymal nodes) or N2 (extra pleural, mediastinal and internal mammary, peri-diaphragmatic, pericardial, and intercostal nodes), while N3 indicated metastases to contralateral intrathoracic and supraclavicular nodes.

After analysing the IASLC database of 2432 cases (1603 with clinical (c) staging, 1614 with pathologic (p) staging, and 785 with both), an important result, already noticed by Rusch et al. [16,17], was confirmed: patients with pN1 or pN2 tumours had worse survival compared to pN0 patients; however, no differences in terms of survival were reported between pN1 and pN2 patients [14].

Hence, among important changes in MPM staging system, in the 8th edition of TNM previous N1 and N2 lymph nodes are gathered into one group (N1) and previous N3 is reclassified as N2 category (Table 1).

A recent study conducted by He et al., analysing 2138 patients extracted from SEER database, has shown that the presence of N2 disease is a worse prognostic factor (HR = 1.608, *p* = 0.001). Moreover, the authors have highlighted that N2 shared equal survival outcome than M1 patients, with only 5 and 6 months of median survival, respectively [18].

## 4. MPM Nodal Drainage Pattern

The nodal drainage pattern of parietal pleura is not well defined. Pre-human models have shown preferential drainage to the mediastinum [19] and post-mortem studies have suggested that diaphragmatic pleural, via the pulmonary ligament and peri-oesophageal tissue, drains into the peri-tracheobronchial lymph nodes [20].

In 2008, Flores et al., analysing a cohort of 348 patients, highlighted that MPM has a greater propensity to metastasize to mediastinal nodes (the most common sites were 4R, 7, and 10R for right-sided tumours and 5, 7, and 10 L for left-sided tumours) and that previously classified as N2 nodes are the first site of drainage from pleura [21]. Moreover, involvement of lymph nodes in unusual locations (e.g., paravertebral, internal mammary, peri diaphragmatic) has frequently been noticed [22].

Notwithstanding that the relevance of extra-mediastinal–intercostal and peri-diaphragmatic groups, nodal involvement is still not clearly defined due to paucity of data [5], an interesting study by Friedberg and co-workers on 56 MPM patients has shown that metastasis to posterior intercostal lymph nodes (ICLN) is associated with poorer survival: ICLN+ patients have a 2.2-fold (*p* = 0.001) and 2.5-fold (*p* < 0.001) increase in risk of death and progression, respectively, compared with ICLN- patients. One of the authors hypotheses is that posterior intercostal LN involvement can be considered as a surrogate for the local spread: ICLN are located outside of the pleural space; thus, spreading beyond the endothoracic fascia would be a sign of regional invasiveness [23].

## 5. How to Stage N?

### 5.1. Clinical versus Pathological Staging

The clinical nodal staging seems to not accurately predict pathologic status: a nodal upstaging was found in 33% of patients who were staged both clinically and pathologically, whereas in 6% a downstaging was discovered. No survival difference has been detected between clinical stages N0, N1 and N2 [14]. The concordance between clinical and nodal staging has been investigated in a large cohort of N+ MPM patients (staged with 7th edition of TNM). c-stage and *p*-stage consistency was found in 59% for cN1, 84% for cN2 and 63% for cN3; while a downstage in 14%, 16% and 37% and upstage in 26%, 0.4% and 0% were reported in cN1, cN2 and cN3, respectively [24].

### 5.2. The Role of Imaging Tests

◦Chest-CT: According to ERS/ESTS/EACTS/ESTRO 2020 guidelines non-invasive imaging is inaccurate in the assessment of nodal involvement: the nodal size and the probability of metastases seem not be correlated [5].

Schouwink et al. performed mediastinoscopy in 43 patients with MPM and compared the staging accuracy with that of CT. Sensitivity, specificity, and accuracy were 80, 100, and 93%, respectively, for mediastinoscopy, versus 60, 71, and 67%, respectively, for CT [25].

The N-staging accuracy of CT has been also reported in a series of 60 resected MPM patients: nodal category was equivalent to histopathology in 27/60 (45%) cases [26].

However, tumour volume can predict nodal stage: the risk of nodal metastases was 14% in patients with tumour maximal thickness of <5.1 mm, while the risk rose to 38% in case of maximal thickness >5.1 mm (*p* < 0.0001) [27].

◦FDG/PET-CT: Due to the close proximity of involved pleura, the power of PET-CT in the assessment of N stage is inadequate: the reported sensitivity in detection of lymph-node involvement is 11% [28].

PET-CT determines N stage with accuracies of 35%, the reported under-staging is 35% and over-staging is 29%. PET-CT has low sensitivity for T4 (67%) and N2 (38%) disease [29].

However, comparing CT alone versus PET-CT, the specificity for stage II and III rises from 77% versus 100% (*p* < 0.01) and 75% versus 100% (*p* < 0.01), respectively [30].

The N-staging accuracy of PET-CT compared to CT alone has reported in the paper by Elliot et al.; comparing preoperative staging with histopathology analysis, the authors have shown a concordance between PET-CT staging and *p*-stage in 38/60 patients (63.3%), superior to that of CT alone (*p* = 0.001) [26].

◦MRI: A prospective clinical trial shows that MRI is not superior to CT regarding detection of lymph node metastases. PET-MRI is as accurate as PET-CT in staging, whereas radiologists felt significantly more confident staging PET-MRI compared to PET-CT using dedicated sequences [31]. Plathow et al. compared the accuracy of the imaging technique in a series of 54 patients: in stage II MPM: the accuracy was 0.8 for MRI comparing to 0.77 of CT and 1.0 of PET-CT; in stage III was 0.9 (MRI) versus 0.75 (CT) and 1.0 (PET-CT) [32].

### 5.3. The Role of Invasive Mediastinal Staging

According to NCCN guidelines, if surgery is considered as a treatment option, the mediastinal lymph nodes should be staged by cervical mediastinoscopy (CM) or by endobronchial ultrasonography (EBUS) [33]. Asco guidelines recommend a mediastinoscopy and/or endobronchial US if enlarged and/or PET+ mediastinal nodes are present, if cytoreductive surgery is planned [6]. ERS/ESTS/EACTS/ESTRO Guideline propose as a possible algorithm a FDG-PET/CT + EBUS/EUS as first step after total body CT to stage patients candidate for surgical resection, and mediastinoscopy for further mediastinal staging in case of borderline resectability [5].

◦Cervical mediastinoscopy (CM)

CM has been reported to have low sensitivity (28–36%) and high false negative rate (22–53%) in MPM staging [34]. This low performance can be explained by the fact that CM cannot reach hilar, internal mammary, intercostal, peri-diaphragmatic and pericardial nodes, which are commonly involved in MPM [14].

Moreover, subcarinal nodes (station 7), that have been reported to be the most common station involved in MPM on invasive staging (36% of patients), have been correctly detected in only 50% on preoperative CM [35]. One of the possible reasons is that CM cannot reach the posteriorly located station 7 nodes, with the only access to the anteriorly and caudally sited station 7 [36]. In addition, while generally safe, a recent meta-analysis shows an overall complication rate of 6%, with a procedure-related mortality of 0.5%, with morbidity (Clavien-Dindo grade III–IV) and laryngeal recurrent nerve palsy reported in 1.9% and in 2.8% of patients, respectively [37].

◦Endobronchial Ultrasound (EBUS)

Rice et al. have analysed the sensitivity and negative predictive value (NPV) of EBUS-TBNA and CM for intrathoracic staging in MPM: the EBUS has a higher sensitivity (58%) compared to CM (28%) with a NPV of 49% vs. 57%, respectively [14].

Although EBUS does not have access to all nodal stations (e.g., internal mammary, peri-diaphragmatic and intercostal areas), it can identify patients with positive upper mediastinal nodes, that have a worse prognosis than that with lower or extra-mediastinal positive nodes [38,39].

Nakas et al. have analysed data on 212 MPM surgically treated patients, dividing the whole cohort into 4 groups according to involved lymph nodes recognised in the surgical specimen: Group 0 (no nodal disease), Group CM (nodes reachable by CM: Stations 2, 3a, 4 and 7), Group EBUS/EUS (nodes reachable by EBUS or EUS: Stations 2, 3a, 4 and 7–11), and Group EM (extra-mediastinal nodes not reachable by CM or EBUS/EUS: Stations 5, 6, internal mammary, pericardial and diaphragmatic lymph nodes). Investigating survival between CM and EBUS subgroups, no significant difference has been detected: the maximum theoretical additional diagnostic yield from EBUS was 26% (30 cases), with a median survival not significantly worse (13.6 months versus 11.3 months for CM and EBUS, respectively). Interestingly, the EM group has a better survival (18.7 months) than that of CM and EBUS (*p* = 0.002) [39].

◦Transcervical Extended Mediastinal Lymphadenectomy (TEMLA) and Thoracoscopy

Surgical invasive staging with transcervical extended mediastinal lymphadenectomy (TEMLA) has been considered for mediastinal staging in MPM patients [40]. Although TEMLA can detect additional involved nodes unseen with EBUS/EUS, the complication (6.0–13.2%) and mortality (1.2%) rate make this technique not frequently used as a pre-treatment patient assessment [5]. Likewise, contralateral thoracoscopy and laparoscopy in nodal staging have been uncommonly applied [41].

## 6. How to Treat N+?

The general aim of surgery in MPM should be the macroscopic complete resection (MCR) of the tumour, which can be reached by either extra pleural pneumonectomy (EPP) or extended pleurectomy/decortication (EP/D) (Table 2) [42,43,44].

There are currently neither prospective trials nor high-volume retrospective studies for patients with clinically node-positive MPM and the management of N+ MPM—particularly the role of surgery—remains extremely debated [24].

According to NCCN guidelines, N1 disease may be managed similar to N0 disease with EP/D and EPP, considered both reasonable surgical options, with complete gross cytoreductive intent in selected patients. Nevertheless, in case of N2 disease, surgery should only be considered in high volume centres with expertise in MPM or in the setting of a clinical trial. Thus, multimodal therapy, palliative therapy, or observation are the treatment option for N2 patients [33].

ERS guidelines state that surgery can be applied to carefully and highly selected MPM patients. Due to the lower respiratory postoperative morbidity rate and the better quality of life maintenance, EP/D should be preferred rather than EPP, as a part of multimodality treatment and in experienced centre. Regarding N2 disease, radical surgery should not be considered except in the context of research [5].

The American Society of Clinical Oncology (ASCO) Clinical Practice Guideline strongly recommend maximal surgical cytoreduction, that includes both EPP and EP/D with lung-sparing option as a first choice, in selected patients with early stage disease. In the case of histologically confirmed contralateral mediastinal or supraclavicular lymph node involvement, neoadjuvant treatment should be administered before consideration of maximal surgical cytoreduction (MSC). Contralateral (N3, 7th edition TNM) or supraclavicular (N3, 7th edition TNM) disease should be a contraindication to MSC. Patients with ipsilateral histologically confirmed mediastinal N+ should only be treated with MSC in a multimodality therapy (neoadjuvant or adjuvant chemotherapy) setting, better if enrolled in clinical trial [6].

Since there is a lack of prospective randomized trials, an evidence-based treatment for N+ MPM patients is still limited. In order to analyse the management of node positive MPM patients, Verma et al. have collected data of 2548 N+ MPM patients from National Cancer Database (NCDB). According to the 7th edition of TNM the cohort was composed by: 20% N1, 70% N2, and 9% N3. Only 328 (13%) of patients were treated with surgery plus chemotherapy (mostly adjuvant, in 6% neoadjuvant), 116 (5%) underwent surgery without chemotherapy, 1204 (47%) received chemotherapy alone, 776 (30%) were observed, and in 124 (5%) treatment information were lost. Among surgically treated cases, 102 (23%) received EPP and 342 (77%) P/D. No survival difference between the two procedures was reported. The median survival of the whole cohort was 9.2 months; the median OS for each subgroup was: 10.0 months (8.7–11.6) for N1, 9.1 months (8.7–9.6) for N2 and 8.5 months (7.0–10.1) for N3. N2+ patients were less likely to be surgically treated and experienced lower OS. The principal conclusion of the authors is that even though more advanced nodal disease is related with a worse prognosis, the nodal status should not be the only exclusion criterion regarding resection: age, performance status and histology should be considered during decision making [24].

Based on data of high-volume centres, the N+ MPM disease has not been considered a contraindication to surgery. Node positive patients with good prognosticators (e.g., epithelioid histology) who were treated with macroscopic resection, showed a high OS [21,23,38].

Data from the SEER database reported a DFS of 18 months and OS of 51 months in pN+ patients surgically treated. Differently, the OS of N+ patients treated with chemotherapy is 14 months, with a median survival of 8 months for patients non surgically treated (4 months with palliative management only) [45].

Conversely, other authors believe that N+ disease should be considered a contraindication to surgery since the advantage of gross resection in patients with poor prognoses is low [46], with a poor survival observed with surgery for N2 (7th edition) disease similar to that reported for pemetrexed/cisplatin chemotherapy without surgery [47].

Data from IASLC database has shown a median survival of 13.4 month for N2 MPM patients (8th edition TNM), with a 24 and 60-month survival of 27%, and 0%, respectively [14].

The role of multimodality treatment in N+ disease is under debate. Data on 60 patients treated with induction chemotherapy followed by EPP and hemi thoracic RT, in a single centre, has revealed that there was no difference in survival of patients with histologically proven N2 (7th TNM) whether all trimodality therapy was completed (12 versus 14 months; *p* = 0.9). Conversely, patients who had no mediastinal node involvement and completed the entire trimodality therapy regimen showed a median survival of 59 months, versus 8 months of patients who did not complete the treatment [46].

In a multicentre phase II trial on 77 patients treated with induction pemetrexed/cisplatin, EPP, and radiation therapy, the median OS of patients with N1 or N2 disease (7th TNM) who completed all therapy was 29.1 months versus 16.6 months of patients who did not complete trimodality regimen [48].

In order to identify patients that benefit the most from multimodality treatment, a cohort of 186 MPM (T1-3, N0-2, M0) patients treated by induction chemotherapy followed by EPP has been studied. A scoring system based on four parameters (tumour volume before chemotherapy >500 mL, non-epithelioid histology, c-reactive protein value > 30 mg/L before chemotherapy, and progressive disease after chemotherapy) with a maximum possible score of 4 and minimum 0, has been proposed. The OS of the whole cohort was 22 months and the Multi-Modality Prognostic (MMP-) Score was a strong independent prognosticator [49].

## 7. Does the Number of Involved Nodes Matter?

As for lung cancer, the number of metastatic nodes and the lymph node ratio (LNR) have been analysed as prognosticator in N + MPM patients. 

The retrospective analysis by Flores et al. on 348 surgical patients reported a significantly worse survival in patients with two or more mediastinal stations (*p* < 0.001) [21].

Moreover, data on 92 MPM patients surgically treated suggests that the number of positive nodes correlates with survival (*p* = 0.001), while the number of involved stations and their location do not [50]. 

Conversely, a study on 99 patients by Hysi et al. has shown that a lower metastatic LNR (≤13%) was related with a significantly improved median survival (19.9 vs. 11.7 months, *p* = 0.01), while OS was not associated to the number of involved or total removed lymph nodes [4].

## 8. How to Predict Tumour Response?

Several parameters related to both macroscopic and microscopic tumour characteristics are involved into MPM progression. Recently, the growing interest in tumour immune microenvironment has revealed that prognostic subgroups of patients can be defined to predict the response to therapies [51].

In order to define which category of patients can benefit the most from a muldimodality approach, several prognostic models have been developed. As previously mentioned, the MMP-score, combining four tumour parameters, has been revealed as a good prognosticator in a cohort of MPM patients [49].

A recently published paper has presented the Mesothelioma Risk Score, a new model which combines pre-treatment parameters gained from imaging (e.g., tumour volume), tumour biopsy (e.g., molecular subtypes) and blood tests (e.g., neutrophil to lymphocyte ratio) to assess the expected outcome of surgery and improve patient stratification [52].

However, despite the recent updates, there has not been a crucial breakthrough in MPM management and further studies are needed to better predict treatment response and select potential surgical candidates.

## 9. Conclusions

Several open questions on staging and management of node positive MPM patients still exist. 

Nodal status is a strong prognostic factor for median survival; however, whether the location or the number of positive nodes is the main prognosticator is still debating. 

Clinical staging alone, including total body CT and PET scan, it is clear to be not sufficient to identify surgically treatable patients. However, even invasive staging with cervical mediastinoscopy and/or endobronchial ultrasound, has shown several limits in MPM. The role of surgery in N+ MPM is highly debated: there is no level I evidence to support surgery for MPM, but these patients are not considered un-resectable. Several patients seem to benefit from a surgery-based approach, particularly those with epithelioid histology, lower-volume disease, and minimal nodal involvement. Anyway, surgery alone seems to not be appropriate and a multimodality approach, preferably as part of a clinical trial, should be considered.

The future debate about the “perfect management of N+ MPM” will be focused more deeply on the open questions that still remain unanswered. Large prospective randomized trials are mandatory to establish an evidence-based approach in order to better stage and treat node positive MPM patients.

## 10. Area for Further Research

Investigate the prognostic value of LNR and number of involved nodes, in N+ patients;Analyse tumour volumetry and assessing its role in staging, treatment response, and as a predictor of OS;Investigate the clinical routine applicability of the proposed circulating biomarkers in diagnosis and as prognosticator;Clarify the definition of “resectable disease” in N+ MPM;Define a prognostic score helping patient allocation for surgery;Determine the role of immunotherapy, new targeted therapies and cancer vaccines in MPM treatment.

## Author Contributions

Conceptualization, S.R., G.C.; methodology, S.R., F.C., M.O.J.; validation, M.D.M., A.R.; formal analysis, S.R., G.C.; investigation, S.R.; data curation, S.R., L.C.; writing—original draft preparation, S.R., G.C.; writing—review and editing, S.R., F.C., M.D.M., M.O.J., A.R., L.C., G.C.; supervision, G.C., A.R.; project administration, S.R., G.C. All authors have read and agreed to the published version of the manuscript.

## Figures and Tables

**Table 1 jcm-10-05177-t001:** N descriptor in MPM by the TNM Classification.

N Category	7th Edition TNM	8th Edition TNM
Nx	Regional lymph nodes not assessable	Regional lymph nodes not assessable
N0	No regional lymph node metastases	No regional lymph node metastases
N1	Metastasis in ipsilateral bronchopulmonary and/or hilar lymph nodes	Metastases to ipsilateral intrathoracic nodes: bronchopulmonary, hilar, or mediastinal lymph nodes (including the internal mammary, peri diaphragmatic, pericardial fat pad, or intercostal lymph nodes)
N2	Metastasis in subcarinal, ipsilateral internal mammary, mediastinal or peri diaphragmatic lymph nodes	Metastases to contralateral intrathoracic nodes (contralateral bronchopulmonary, hilar, or mediastinal lymph nodes) or ipsilateral or contralateral supraclavicular lymph nodes
N3	Metastases in contralateral mediastinal, contralateral internal mammary, or hilar lymph nodes and/or the ipsilateral supraclavicular or scalene lymph nodes	/

**Table 2 jcm-10-05177-t002:** Definition of surgical procedures according to IASLC [42] and ERS/ESTS/EACTS/ESTRO Guidelines [5].

Surgical Procedure	Definition
**EPP** Extra Pleural Pneumonectomy	en-bloc resection of the parietal and visceral pleura with the ipsilateral lung, pericardium, and diaphragm combined with systematic mediastinal lymph node dissection
**P/D** Pleurectomy/Decortication	parietal and visceral pleurectomy and removal of all gross tumour without removal of diaphragm and pericardium
**EP/D** Extended P/D	parietal and visceral pleurectomy to remove all gross tumour with resection of the diaphragm and/or pericardium
**(p)P/D** Partial P/D	partial removal of parietal and/or visceral pleura

## Data Availability

No new data were created or analysed in this study. Data sharing is not applicable to this article.

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
