# Peer review of "Malignant Pleural Mesothelioma Nodal Status: Where Are We at?"

_jcm, 2021, doi:10.3390/jcm10215177_

Round 1

Reviewer 1 Report

Dear authors, I think this is a very good review of an important area, and one that is usually brushed aside, in countries such as the UK, where surgery is not done for MPM. 

I cannot fault anything about the article as it is well written, does not grammatical errors and flows well. 

My only minor point would be for the authors to write a few bullet points as to what they think areas for further research/study should be.....

Well done again

Author Response

Many thanks for your suggestions and comments!

I have added five bullet points at the end of the text.

"Area for further research:

  • Investigate the prognostic value of LNR and number of involved nodes in N+ patients
  • Analyse tumour volumetry and assessing its role in staging, treatment response, and as a predictor of OS
  • Investigate the clinical routine applicability of the proposed circulating biomarkers in diagnosis and prognostic stratification
  • Clarify the definition of “resectable disease” in N+ MPM
  • Determine the role of immunotherapy, new targeted therapies and cancer vaccines in MPM treatment"

Reviewer 2 Report

Dear authors,

Thank you very much for the opportunity to read this Review article on a nodal status in malignant pleural mesothelioma patients. This indeed remains one of the challenges in the treatment of MPM.

Overall, this is a nice overview and summary of the current state of the art as well as the guidelines. Nevertheless, a more accentuated exploration with interpretation of the main challenges respectively pitfalls that arise in setting up the perfect treatment approach for these delicate patients would be desirable.

In one of the last paragraphs the MMP-score is mentioned, in my opinion out of context, although better patient selection should still be discussed in this Review, but in a greater extent and further supporting scores/Tools etc.

Q1:

Running title is a bit misleading. Methods section is not a classical method section. It contains literature overview, interpretation and conclusion respectively discussion. Would separate it more clearly.

Q2:

Correct writing of (e)P/D is according to the newest ERS/ESTS guidelines EP/D.

Q3:

Tiping error line 172 (…, it can be identify…). Cross out "be".

Q4:

Why did you not use the newest guidelines in Table 2? ERS/ESTS…?

Q5:

Please define multimodality therapy approach.

Q6:

Typing error in line 264 . .. to identify patients…

Author Response

I would like to thank you for your comment and suggestions.

Q1: running title has been modified and sections have been revised

Q2: done

Q3: corrected

Q4: updated

Q5: added, lines 50-52

Q6: corrected

Reviewer 3 Report

It is my great pleasure to review this manuscript. This review proposed several important issues which are still controversial. The authors tried to focus on nodal status in MPM, for instance, how to stage N and how to treat N+ patients? This manuscript tried to cover these important topics by analyzing the existing relevant literature on lymph node status in MPM patients.

The manuscript is well-prepared.

Author Response

I would like to thank the Reviewer for the comment.